# Pace Controlled by a Steady-State Physiological Variable Is Associated with Better Performance in a 3000 M Run

**DOI:** 10.3390/ijerph18157886

**Published:** 2021-07-26

**Authors:** Claire A. Molinari, Pierre Bresson, Florent Palacin, Véronique Billat

**Affiliations:** 1Unité de Biologie Intégrative des Adaptations à l’Exercice, Université Paris-Saclay, Univ Evry, 91000 Evry-Courcouronnes, France; contact@billatraining.com; 2BillaTraining SAS, 32 Rue Paul Vaillant-Couturier, 94140 Alforville, France; bresson.pierre91@gmail.com (P.B.); palacinflorent@gmail.com (F.P.)

**Keywords:** V̇O_2max_, performance, running, self-pacing, skewness, steady state

## Abstract

This paper aims to test the hypothesis whereby freely chosen running pace is less effective than pace controlled by a steady-state physiological variable. Methods Eight runners performed four maximum-effort 3000 m time trials on a running track. The first time trial (TT1) was freely paced. In the following 3000 m time trials, the pace was controlled so that the average speed (TT2), average $\dot{V}$O_2_ (TT3) or average HR (TT4) recorded in TT1 was maintained throughout the time trial. Results: Physiologically controlled pace was associated with a faster time (mean ± standard deviation: 740 ± 34 s for TT3 and 748 ± 33 s for TT4, vs. 854 ± 53 s for TT1; *p* < 0.01), a lower oxygen cost of running (200 ± 5 and 220 ± 3 vs. 310 ± 5 mLO_2_·kg^−1^·km^−1^, respectively; *p* < 0.02), a lower cardiac cost (0.69 ± 0.08 and 0.69 ± 0.04 vs. 0.86 ± 0.09 beat·m^−1^, respectively; *p* < 0.01), and a more positively skewed speed distribution (skewness: 1.7 ± 0.9 and 1.3 ± 0.6 vs. 0.2 ± 0.4, *p* < 0.05). Conclusion: Physiologically controlled pace (at the average $\dot{V}$O_2_ or HR recorded in a freely paced run) was associated with a faster time, a more favorable speed distribution and lower levels of physiological strain, relative to freely chosen pace. This finding suggests that non-elite runners do not spontaneously choose the best pace strategy.

## 1. Introduction

In real races, athletes achieve the best performances by varying their pace [1]. Despite the obvious importance of pacing and the impact of pacing mistakes on athletic performance, there have been few systematic studies of how various pacing strategies influence the outcome of competitive results [2]. With a view to optimizing athletic performance, various self-pacing strategies have been used to vary the running speed and minimize the overall time between the start and the finish. Therefore, in order to avoid becoming over-fatigued before reaching the finishing line, the athlete can choose to spontaneously modulate his/her pace during the race [3]. Over the last 20 years, researchers have hypothesized that human athletes regulate their effort during competitive events according to when they expect the exercise bout to end [4,5,6]. In order to maintain homeostasis, the central governor supposedly receives feedback from a receptor that monitors the physiological response to the exercise activity’s demands [5,7,8,9]. In line with the “central governor” paradigm of health and performance, free pacing is currently considered to be the key to better performance [4]. In a race, the choice of the speed variations that will maximize the endurance athlete’s ability to win a race involves complex interplay between physiological and psychological factors [10].

A recent analysis of almost 300 marathon races showed that positive speed asymmetry was associated with better performance [11]. One particular example of positive speed asymmetry is the bathtub-shaped curve, in which the runner spends two-thirds of the duration of the race below his/her overall average speed (i.e., after a fast start and before a fast finish). In cases of negative asymmetry, the runner spends much more time (and thus runs further) above his/her overall average speed [11]. It is possible to gain a better understanding of running performance by combining personalized and generalized analyses. Hence, along with “big data” analyses of pacing strategies, it is essential to examine conventional physiological variables in ecological situations that mirror track events.

Middle- and long-distance runs (such as the 3000 m) push human athletes to their endurance limits. The athlete’s various physiological systems interact to regulate the running speed and maintain homeostasis as long as possible [5]. Indeed, simple physiological markers, heart rate (HR) and oxygen uptake ($\dot{V}$O_2_) are involved in competitive efforts, so that the burden of fatigue is proportional to the distance completed.

Therefore, the objective of the present study was to determine how an imposed, constant $\dot{V}$O_2_ or HR (corresponding to the mean value measured in a free-paced trial) would influence overall performance, physiological strain, speed variations, and the speed distribution (asymmetry) over a 3000 m run.

We hypothesized that a non-elite runner can maximize performance and minimize physiological strain by using a steady-state physiological parameter ($\dot{V}$O_2_ or HR) to control pace and maintain homeostasis for as long as possible during an effort (a 3000 m run).

## 2. Materials and Methods

Eight male, non-elite runners (mean ± standard deviation (SD) age: 36.5 ±11.8; weight: 69.8 ± 4.5 kg; height: 176.1 ± 2.8 cm) volunteered to participate in the study. All runners had been carrying out a regular training volume of three to four times per week for more than 5 years. The study’s objectives and procedures were approved by an institutional review board (CPP Sud-Est V, Grenoble, France; reference: 2018-A01496-49). All participants were provided with study information, and gave their written consent to participation.

Participants individually performed four different 3000 m time trials. All trials were performed at the same time of day (±2 h), with a recovery time of between 3 and 7 days between trials. Participants were asked to refrain from training during the 24 h preceding each test. All four trials were performed on an officially measured outdoor athletics track.

### 2.1. Trial Conditions

The first 3000 m time trial (TT1) was freely paced; all participants were instructed to run as fast as possible at a self-selected pace. The total trial time was measured using a stopwatch. TT1 was the reference trial from which the control variables applied to the following three trials (performed in random order) were determined. For the statistical analysis, we had classified the results as follows: In the second 3000 m time trial (TT2), the runner had to maintain a steady-state corresponding to the average speed measured in TT1. In the third time trial (TT3), the runner had to maintain a steady-state $\dot{V}$O_2_ corresponding to the average value recorded during TT1. In the fourth time trial (TT4), the runner had to maintain a steady-state HR corresponding to the average value recorded in TT1. The time trials were preceded by a 15 min at a low speed (about 50% of $\dot{V}$O_2max_) warm-up period on the track, following at 3 min session of “crossing” at increasing speeds. After this, the runners were equipped with the K5 harness.

### 2.2. Experimental Measurements

Respiratory gases (oxygen uptake ($\dot{V}$O_2_) and carbon dioxide production ($\dot{V}$CO_2_)) were continuously measured using a telemetric, portable, breath-by-breath sampling system (K5, Cosmed, Rome, Italy) [12]. A global positioning system watch (Garmin, Olathe, KS, USA) paired with the K5 system was used to measure the HR and the speed response (using 5 s data averages) throughout each trial.

The K5 long-range telemetry system was used to monitor the steady states in TT3 and TT4. The investigator (in the center of the track) received the data from the K5 and the Garmin system on a laptop computer in real time. He/she could then tell the runner to increase, decrease or maintain his speed in order to maintain the steady state. The runner was able to monitor his instantaneous speed using the watch. In TT2, the watch had been programmed to give sound alerts (beeps) if the speed deviated by more than 5%. The investigator also monitored the running speed by checking the K5 telemetry system.

### 2.3. Calculated Variables

*Oxygen cost* (mLO_2_·kg^−1^·m^−1^) was calculated as the ratio between oxygen consumption (mLO_2_·kg^−1^·min^−1^) and speed (m/min).

The *cardiac cost* (m^−1^·beat) of running was calculated as the ratio between the 5-s-averaged running speed and the 5-s-averaged HR. This cost ratio corresponds to the distance (m) covered at each beat [13].

### 2.4. Statistical Analysis

All statistical analyses were performed using XLSTAT software (version 2019.1.1, Addinsoft, Paris, France). All the test variables ($\dot{V}$O_2_, $\dot{V}$CO_2_, HR, and speed) were reported as the mean ± SD. For each variable, the normality and homogeneity of the data distribution were examined in a Shapiro–Wilk test. Paired t tests were used to examine mean differences between outcome variables (race time, percentage of time spent below average speed (i.e., positive asymmetry), and the cardiac cost). Kendall’s τ was calculated in order to detect time trends in physiological and speed responses during the four trials. Furthermore, we calculated the skewness value of the speed distribution as (mean − median)/SD. The skewness is a measure of the asymmetry of the probability distribution of a real-valued random variable about its mean. Its value can be positive, negative, or undefined: a positive skew means that the mean is greater than the median, while a negative skew means the mean is less than the median. We compared the time spent below the average speed with the time spent above the average speed. Positivity asymmetry was defined as a run with 54% or more of the time spent below the average speed, and negative asymmetry was defined as a run with 46% or less of the time spent above the average speed [11].

## 3. Results

### 3.1. Performance over 3000 m in Each Time Trial

The completion times in the four 3000 m time trials are summarized in Table 1. The completion time was significantly lower in TT3 and TT4 (when pace was based on a constant $\dot{V}$O_2_ or HR, respectively) than in the freely paced TT1 (*p* = 0.009 and *p* = 0.002, respectively). In contrast, the time in TT2 (run at constant speed) did not differ significantly from that in TT1 (*p* = 0.098). We conclude that a physiologically controlled pace strategy is associated with better 3000 m running performance.

### 3.2. Physiological Strain: Oxygen and Cardiac Costs

The physiological strain (oxygen and cardiac costs, in mLO_2_ per meter run and meter run per beats, respectively) was significantly lower in TT3 and TT4 than in the freely paced TT1 (*p* = 0.015 and *p* = 0.001, respectively; Table 1).

### 3.3. Statistical Characteristics of Speed Distribution Strategies: Skewness, Coefficient of Variation, and Kendall’s τ

The speed distribution was more positively skewed in TT3 and TT4 (respectively, 1.7 ± 0.9 and 1.3 ± 0.6 for the skewness, which is a dimensionless number) than in TT1 (0.2 ± 0.4; *p* < 0.05) (Figure 1). To determine the asymmetric nature of the speed distribution, we calculated the percentage time spent above the runner’s average speed and percentage time spent below.

All the runners displayed positive speed asymmetry, meaning that they tended to run for longer below their average speed (Table 1). The percentage time spent below the average speed was significantly greater when the 3000 m pace was run at constant $\dot{V}$O_2_ or HR (TT3 or TT4, respectively) than in the free pace trial (TT1) (*p* = 0.025 and *p* = 0.010, respectively).

The speed coefficient of variation was significantly greater in TT1 than in TT4 (14.6 ± 2.5 vs. 7.5 ± 2.7%, respectively; *p* = 0.001). However, there was no significant difference between TT1 and TT3 (14.6 ± 2.5 vs. 9.9 ± 5.1, respectively; *p* = 0.077).

We used Kendall’s τ to characterize the time trend in the speed during each trial. We observed a decreasing trend: the mean ± SD Kendall’s τ was −0.273 ± 0.12 for TT1, −0.499 ± 0.22 for TT2, −0.138 ± 0.25 for TT3, and −0.507 ± 0.24 for TT4. Given that all the p-values were below 0.05, we rejected the null hypothesis stating that the speed time trend was constant. The negative Kendall’s τ values indicate that the runners slowed progressively during the time trial, regardless of the instructions or controlling variables (Figure 2).

## 4. Discussion

Our present results confirm the hypothesis whereby the imposition of a stable physiological state (for oxygen consumption or HR) optimizes the speed variation pattern. Indeed, the participating runners were significantly faster over 3000 m when the pace was controlled by a constant HR or a constant $\dot{V}$O_2_. Hence, our results confirmed the hypothesis whereby a spontaneously chosen pace does not enable the non-elite runner to maximize performance or minimize physiological strain. This is in accordance with Katch et al.’s criticism of the widely used Cooper 12 min running endurance test, which was widely used in 1973 [14,15]. The researchers suggested that the Cooper test penalizes less experienced runners because the alternation of running and walking was possible, and the choice of pace was left to the runner [15]. Katch et al. [15] concluded that further experiments were needed to determine optimal pacing requirements and test durations for endurance running performance.

In the present experiment, we controlled the running pace with reference to $\dot{V}$O_2_ or HR. We found that free pacing was not optimal in non-elite runners. Indeed, the runners performed better and experienced lower levels of physiological strain (oxygen and cardiac costs) when the 3000 m time trial was run at a constant $\dot{V}$O_2_ or a constant HR (in TT3 and TT4, respectively), relative to the freely paced trial. Furthermore, the runners displayed a positively asymmetric speed distribution and a lower coefficient of variation for speed in TT2, TT3, and TT4. Indeed, the literature data show that positive speed asymmetry and a lower coefficient of variation for speed are associated with better race performance [11].

Historically, the most popular method for observing physiological responses during exercise has been the constant load or fixed-intensity protocol; this is based mainly on the assumption that an organism has a certain threshold response under a given condition. However, constant-load exercise does not wholly allow for haphazardness or variability, since the biological system is overridden by a predetermined, externally imposed load that cannot be altered [16]. Bath et al. [17] showed that a second runner had no effect on the pacing strategy, running speed, HR or the rating of perceived exertion during a 5 km time trial—even though the athletes thought that their performance had been improved. Hence, Bath et al. [17] considered that an athlete’s subconscious pacing strategy is robust and is not altered by the presence of another runner. However, Konings et al. [18] recently showed that the behavior of an opponent affects the decision-making process and pacing behavior of an athlete when competing in a race. Athletes were able to handle higher levels of peripheral fatigue [19]. In contrast, in competitive sports such as middle-distance and marathon running, sub-elite runners tend to adopt a pacing strategy in the beginning of the race that they cannot sustain until the end of the race [19,20]. That is why we decided to determine whether a physiological steady-state variable could improve performance by forcing the runner to adopt a better “strategy” and perhaps modify his/her speed distribution.

It may now be opportune to explore an ecological approach to exercise limitation by adopting a mechanical control variable (e.g., power, speed, or cadence) or a physiological control variable (HR or $\dot{V}$O_2_). In the context of middle-distance running (races ranging from over 1500 to 10,000 m, lasting between 4 and 15 min), speed is not constant; we still do not know how a runner regulates his/her speed to achieve optimal performance and to avoid exhaustion before the end of the race [21]. If one continues to reason along these lines, the use of $\dot{V}$O_2_ or HR as a controlling variable might induce an ideal pacing pattern (or at least one that is better than freely chosen pace) [1].

Indeed, compared with a constant-pace run, a freely paced run enabled athletes to finish with a sprint; however, the sprint increased performance but was not perceived as being less strenuous [22]. Therefore, according to Foster et al. [23], there is some evidence to suggest that the initial speed and anaerobic energy use was high, and decreased to a more or less constant value over the remainder of the race. This apparent monitoring of the metabolic disturbance vs. the demands of an exercise bout has recently been integrated into a cohesive “teleoanticipation” or “central governor” hypothesis [7,9,21]. Therefore, the physiological limits on exercise are frequently studied in situations where: (i) the subject is free to regulate his/her pace [1,21,24]; and (ii) the relationships between speed variations and physiological signals are known. We previously reported that the coefficient of variation for speed ranged from 1% to 5% in 3000 to 10,000 m events for competitive runners [25], 3% in marathons [26], and 5.4% for the top ten finishers in a 100 km ultramarathon [27]. The coefficient of variation for speed for a recent marathon winner (Eliud Kipchoge) was 2.2%, which testifies to a relatively even pace strategy [28]. However, the asymmetry of speed distribution must also be explored.

Maron et al. [29] suggested that speed variations could enable a runner to reach $\dot{V}$O_2max_, and so a marathon should not be run at a constant speed. Indeed, the researchers reported that a rather good runner (marathon time: 2 h and 36 min) reached 100% of his $\dot{V}$O_2max_ at a distance of 23.4 miles. The researchers also indicated that the oxygen uptake was between 68% and 100% of $\dot{V}$O_2max_. For middle-distance running (800–3000 m) specifically, it has been shown that $\dot{V}$O_2max_ is always achieved [30,31,32].

Our second starting hypothesis was that the physiological strain might be lower in a physiologically controlled time trial. Indeed, we found that the oxygen and cardiac costs [13,33] were lower in TT2 and TT3 than in TT1. It has long been known that the oxygen cost of running is a key factor in endurance performance, along with $\dot{V}$O_2max_ and the ability to sustain $\dot{V}$O_2max_ at the critical speed or in a peak lactate steady state [30,34].

It is well established that the cardiac cost (the relationship between the HR and running speed, measured here in m·min^−1^) changes during high-intensity continuous exercise. At a constant running speed, the HR increases over time. In contrast, maintenance of a constant HR means that the running speed must fall over time [13]. In real circumstances, athletes generally know the distance of the competition and thus choose the optimal running speed accordingly. They frequently exploit the HR response to regulate their running intensity and optimize their performance; the objective is often to maintain a constant HR [13]. In a laboratory-based study of the change over time in the HR-speed relationship, Boudet et al. [13] simulated races at four different high running speeds (ranging from 82–90% of v$\dot{V}$O_2max_, and lasting between 36 and 9 min). By continuously monitoring the HR and the speed, they calculated a novel index: the HR-running speed ratio, i.e., the distance (in m) covered for each beat [13]. The researchers showed that when the speed was constant, cardiac drift was observed even in short (10 min) runs at 90% of v$\dot{V}$O_2max_, and thus led to a fall in the HR-running speed ratio. By analogy with the oxygen cost (a familiar concept that is reportedly a key factor in long-distance running performance), Boudet et al. [13] referred to the HR-running speed ratio as the cardiac cost [35]. In a study of a marathon run, the cardiac cost increased throughout the race [36]. In the present study, we observed the same range of cardiac cost values as Boudet et al. did [13], although our protocol set the HR to be the speed-controlling variable (i.e., an independent variable, instead of a dependent variable at constant speed). Even though few coaches have access to a portable gas analyzer system, we suggest that HR control might be a means of increasing performance and (perhaps) positive speed asymmetry.

Our study reveals that physiological controllers helped male runners to improve performance. However, even if the protocol was designed to enable the runner to achieve the best possible performance, the magnitude of improvement with subsequent trials could suggest that runners’ performance (since they are non-elites) could have been different with repeated attempts of TT1. Even though $\dot{V}$O_2_ and HR were maintained in a steady state, we could not determine whether or not the runner was in homeostasis [37]. We can nevertheless say that the runners were exhausted at the finish, given that their RPE was 19–20 at the end of TT1, and that the speed decreased (according to Kendall’s τ). The best performance was obtained when paced was controlled by $\dot{V}$O_2_.

In a constant power paradigm, the changes in pulmonary gas exchanges (and especially $\dot{V}$O_2_) during constant-load exercise have been used to define three intensity domains: moderate, heavy, and severe [34,38,39,40]. Next, the $\dot{V}$O_2_ steady state (i.e., $\dot{V}$O_2max_) is used to differentiate between subcritical and supracritical exercise intensities [38]. However, our trial design did not follow a constant exercise model, and we were able to obtain a $\dot{V}$O_2_-steady state during maximal, freely paced exercise for more than 10 min. Our present result confirms an earlier report in which well-trained middle-distance runners were able to maintain a steady state at 93% of $\dot{V}$O_2max_ during a 12 min time trial [38].

The constant power model (such as the power eliciting $\dot{V}$O_2max_ in an incremental protocol) is still useful for predicting middle- and long-distance running performance [41] but cannot provide any information about the speed profile—even though it has been shown that the marathon is run at around the critical speed [42]. The concept of skewness and the “bathtub-shaped” pace curve may be of value for pacing in the marathon [11]. Indeed, Billat et al. [11] recently showed that along with conventional speed variables (the average speed and the coefficient of variation), the asymmetry of the speed distribution (skewness) differentiated between elite and non-elite marathon runners [11]. The present study of a 3000 m time trial showed that it was possible to increase performance and come closer to the “bathtub” pace curve by controlling speed with a steady-state physiological variable ($\dot{V}$O_2_ or HR). The study also had limitations. During the steady state trials, the speed variations were performed by the investigator/runner pair since the investigator could report the deviation from the constant to the runner. This could therefore have led to an interpretation of the set point by the experimenter. On the other hand, it would have been interesting to repeat a freely paced time trial at the end of the experiments. Indeed, it is to be expected that athletes’ performance will improve as a result of task familiarization with repeated attempts, which means it is difficult to determine how much of the improvement observed was due to our intervention and how much was due to participants becoming more familiar with the activity. We still do not know how speed variation is integrated into the runner’s strategy, and whether it is a controlling variable or dependent variable (or both)—especially when the runner is close to his/her $\dot{V}$O_2max_.

## 5. Practical Applications

Most runners train by monitoring their heart rate and running speed and the runners or their coaches then define training zones with regard to v$\dot{V}$O_2max_ or HRmax. The steady-state $\dot{V}$O_2_ or HR (at the average value recorded in a freely paced 3000 run) used in this protocol might constitute a relevant tool for producing a training plan suited to the runner’s physiological strain. Moreover, a runner performing alone will be able to monitor his/her heart rate (using a monitor) which will allow him to check the correspondence between the perceived intensity and that physiologically controlled pace.

## 6. Conclusions

The imposition of a steady-state $\dot{V}$O_2_ or HR (at the average value recorded in a freely paced 3000 m run) enabled non-elite runners to improve their performance, achieve a more favorable speed distribution, and experience less physiological strain (relative to the freely paced run). This finding may suggest that good but non-elite middle-distance runners do not spontaneously choose the best pace strategy with regard to performance and physiological strain. Controlling speed with a physiological variable was associated with better performance and a lower level of physiological strain.

## Figures and Tables

**Figure 1 ijerph-18-07886-f001:**
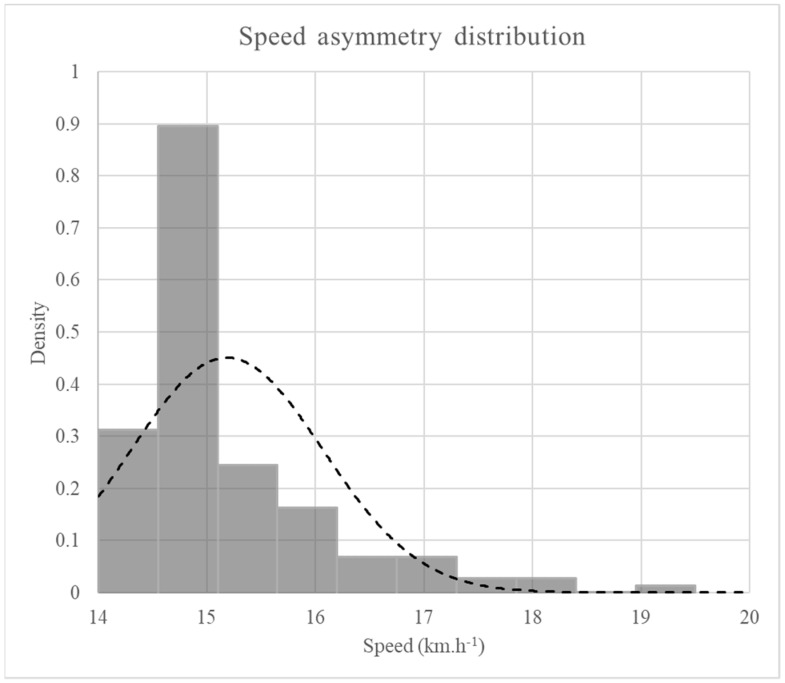
An example of a right-/positively skewed speed distribution (the bar chart) for participant 1 in the TT3 3000 m time trial.

**Figure 2 ijerph-18-07886-f002:**
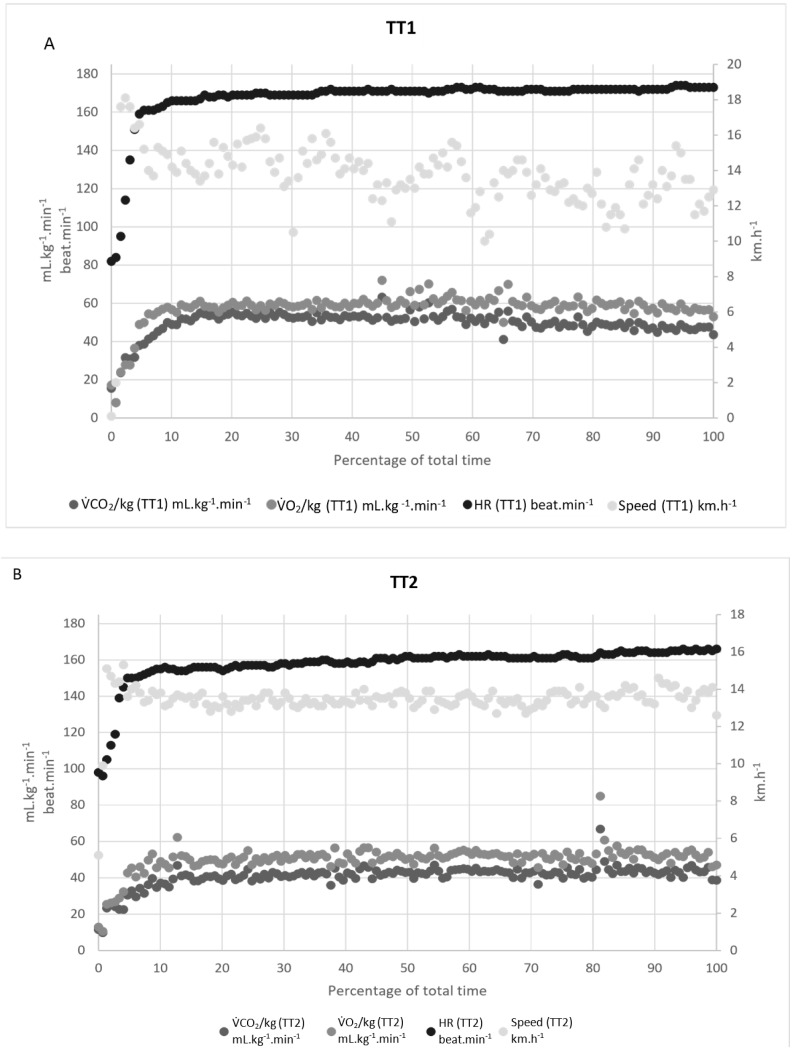
Representative data from participant #1, showing respiratory gas exchange (oxygen uptake ($\dot{V}$O_2_) and carbon dioxide production ($\dot{V}$CO_2_)), HR, and speed over the duration (in %) of the 3000 m time trials. (**A**): a freely paced run (TT1), (**B**): constant speed, set to the average from TT1 (TT2), (**C**): speed controlled by the steady-state $\dot{V}$O_2_ from TT1 (TT3), (**D**): speed controlled by steady-state HR from TT1 (TT4).

**Table 1 ijerph-18-07886-t001:** Summary of the participants’ performance and physiological strain.

Trial	Race Time (s)	Percentage Time Spent Below Average Speed	Oxygen Cost (mLO_2_·kg^−1^·m^−1^)	Cardiac Cost (m^−1^·Beat)
3000 m, freely paced (TT1)	853.6 ± 52.5	51.9 ± 3.8	0.31 ± 0.05	1.17 ± 0.11
3000 m, steady-state speed (TT2)	798.5 ± 60.4	57.1 ± 6.6 ^a^	0.22 ± 0.03 ^a^	1.45 ± 0.12 ^a^
3000 m, steady-state $\dot{V}$O_2_ (TT3)	739.7 ± 34.2 ^a^	64.5 ± 8.3 ^a^	0.20 ± 0.05 ^a^	1.46 ± 0.15 ^a^
3000 m, steady-state HR (TT4)	747.9 ± 32.6 ^a^	61.1 ± 6.0 ^a^	0.22 ± 0.03 ^a^	1.45 ± 0.09 ^a^

Note: ^a^ indicates a significant difference (*p* < 0.05) vs. the freely paced 3000 m time trial (TT1). Positive speed symmetry was defined as a percentage time spent below the average speed of >54%. The data are quoted as the mean ± SD.

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
