# Peer review of "Pace Controlled by a Steady-State Physiological Variable Is Associated with Better Performance in a 3000 M Run"

_ijerph, 2021, doi:10.3390/ijerph18157886_

Round 1
Reviewer 1 Report
It is an interesting and topical research. I congratulate the authors for the great work. I point out some considerations:
- Introduction section:
- It is well planned,
- Lines 57-61 should be removed or become part of methods section
- Methods section
- Indicate type of study
- Indicate physical exercise characteristics of the participants (sports experience, weekly training time, etc.)
- Results section
- It is well planned.
- Discussion section:
- It is well planned
- The last paragraph of the discussion should indicate the limitations of the study.
Author Response
Thank you for your kind remarks, your careful review and for raising these important points.
- Introduction section:
- Lines 57-61 should be removed or become part of methods section
We have removed lines 57-61
- Methods section
- Indicate type of study
- Indicate physical exercise characteristics of the participants (sports experience, weekly training time, etc.)
We have indicate lines 70 to 72.
- Discussion section:
- The last paragraph of the discussion should indicate the limitations of the study.
We have added a summary of the main limitations of the study Lines 298 to 306.
Reviewer 2 Report
The manuscript is quite interesting and add some knowledge and practical ideas to the field of training and pacing during races. Nevertheless, I want to suggest the following issues to the authors:
Methods
The warm up does not appear in the "Trial conditions" section. Maybe it is important to know how and when the participants performed it to better understand the dynamics of HR and VO2.
In the “Calculated variables” section, oxygen and cardiac cost are defined, although in table 1, another variable appears: running cost. Are running cost and oxygen cost the same variable? Please, clarify.
In the same table 1, there are no units to describe Running cost and Cardiac cost. I think they are mLO2/kg/m and m/beat. Is it correct? If they appear in the table, the understanding of it would be highly improved.
In section "3.2 Physiological strain", oxygen and cardiac costs are expressed as mLO2 and beats per meter run, respectively; but in the “Calculated variable” section are defined as “This cost ratio corresponds to the distance (m) covered at each beat”. Please, clarify.
Results:
Regarding my first issue about warm up, in table 2 we can observe how during the first 10% of the race (74-85 seconds) in TT3 and TT4 the initial speed is very high until de VO2 and HR gets the steady state. Maybe with a different warm up, the variables are a bit higher at the beginning, and the steady state are reached earlier, affecting the time of the trial.
Spell mistakes:
Line 139 Milliliters appears as ml. Please, change to mL.
Line 179 Please, change 3000m to 3000 m (spaced).
Line 306, change “tomonitor” to “to monitor”
Author Response
Thank you for your careful review and for raising these important points. We now emphasize the novelty the clarifications and corrections that you have suggested
- Methods
The warm up does not appear in the "Trial conditions" section. Maybe it is important to know how and when the participants performed it to better understand the dynamics of HR and VO2.
We have added the warm-up conditions in section 2.1 lines 89 to 91.
- In the “Calculated variables” section, oxygen and cardiac cost are defined, although in table 1, another variable appears: running cost. Are running cost and oxygen cost the same variable? Please, clarify.
- In the same table 1, there are no units to describe Running cost and Cardiac cost. I think they are mLO2/kg/m and m/beat. Is it correct? If they appear in the table, the understanding of it would be highly improved.
You are right, we have clarified that point in Table 1
- In section "3.2 Physiological strain", oxygen and cardiac costs are expressed as mLO2 and beats per meter run, respectively; but in the “Calculated variable” section are defined as “This cost ratio corresponds to the distance (m) covered at each beat”. Please, clarify.
You are right, we have clarified lines 142-143.
- Results:
Regarding my first issue about warm up, in table 2 we can observe how during the first 10% of the race (74-85 seconds) in TT3 and TT4 the initial speed is very high until de VO2 and HR gets the steady state. Maybe with a different warm up, the variables are a bit higher at the beginning, and the steady state are reached earlier, affecting the time of the trial.
Indeed, your remark is relevant, so we have added to the manuscript the type of warm-up performed by the runners in section 2.1 lines 89 to 91.
- Spell mistakes:
Line 139 Milliliters appears as ml. Please, change to mL.
Line 179 Please, change 3000m to 3000 m (spaced).
Line 306, change “tomonitor” to “to monitor”
Spell mistakes have been corrected in the manuscrit :
Line 142 : change to mL.
Line 182 : 3000 m (spaced).
Line 317, change to “to monitor”.